# Comparison between the Choices Five-Level Criteria and Nutri-Score: Alignment with the Dutch Food-Based Dietary Guidelines

**DOI:** 10.3390/nu14173527

**Published:** 2022-08-26

**Authors:** Jaimy J. C. Konings, Herbert Smorenburg, Annet J. C. Roodenburg

**Affiliations:** 1Department of Nutrition and Health, HAS University of Applied Sciences, 5200 MA ‘s-Hertogenbosch, The Netherlands; 2Choices International Foundation, 7084 AV Breedenbroek, The Netherlands

**Keywords:** food environment, The Netherlands, Wheel of Five, front-of-pack labeling, nutrient profiling, Nutri-Score, Choices five-level criteria, NEVO database

## Abstract

The current food environment in The Netherlands is considered obesogenic. Eighty percent of the products in supermarkets are unhealthy. The Wheel of Five is the well-established, science-based Dutch food-based dietary guideline (FBDG) developed to stimulate healthier choices. In addition, simple directions on food packaging, such as front-of-package (FOP) health logos, could also be helpful. However, these tools should be in line with each other, in order not to confuse the consumer. To study this, we evaluated two FOP nutrient profiling systems (NPSs) for their alignment with the Wheel of Five: Choices five-level criteria and Nutri-Score. For this, a small but representative sample of 124 products was selected from the Dutch food composition database (NEVO). For these products, the scores for Choices and Nutri-Score were calculated using the published criteria, while compliance with the Wheel of Five was established by using the criteria from Netherlands Nutrition Center (NNC). The Wheel of Five food groups were used to categorize the products. Differences between the Wheel of Five and Choices are smaller than with Nutri-Score, concluding that Choices is more consistent with the Wheel of Five and might be an attractive alternation for a FOP health logo on the Dutch market.

## 1. Introduction

Living a healthy life is not only important for your daily life now, but also for your body’s ability to prevent, fight, and recover from infections and chronic diseases. Good nutrition and healthy diets can reduce non-communicable diseases (NCDs) and are important for supporting immune systems [1,2]. The World Health Organization (WHO) calls for comprehensive and integrated action at the country level, led by governments, using existing knowledge and solutions that are highly cost-effective [3]. Solutions include food-based dietary guidelines (FBDGs) and front-of-pack (FOP) labeling. FBDGs are established in many countries in order to serve as a foundation for the public and to promote healthy eating habits and lifestyles [4]. FOP labeling contains a nutrient labeling aspect that allows consumers to make quick decisions regarding a product’s health using a simple and easily readable style [5,6]. Furthermore, it encourages the industry to make healthier products by reformulating them [7,8]. According to WHO, FOP labeling needs to be guided by five guiding principles, with the first principle stating that “The FOP labeling system should be aligned with national public health and nutrition policies and food regulations, as well as with relevant WHO guidance and Codex guidelines” [3]. FOP labeling systems that are currently in practice or preparation in Europe have been compared by Van der Bend and Lissner [9]. They provide a “Funnel Model” to characterize methodological differences between FOP labels, which helps this study to understand the underlying differences and similarities between the FOP labeling systems under consideration.

In The Netherlands, the current food environment is often referred to as obesogenic, because it leads to unhealthy behavior [10]. About 80% of the products in Dutch supermarkets in 2021 did not comply with the established guidelines from The Netherlands Nutrition Center (NNC) [11]. In the same year, half of the adult Dutch population was overweight [12]. The Dutch FBDGs were derived based on twenty-nine systematic reviews that summarized randomized controlled trials, prospective cohort studies on nutrients, food and dietary patterns, and the risk of the top ten major chronic diseases in The Netherlands. Dietary guidelines were formulated for foods and food patterns that led to health gains for those food groups for which there was convincing or plausible evidence [13]. In a follow-up process, these dietary guidelines, current Dutch consumption patterns, and dietary reference values were used in an optimization model to derive the FBDGs for The Netherlands, visualized in the Wheel of Five [14]. The Wheel of Five is based on five food groups: fruits and vegetables; fat sources; protein sources; carbohydrate and fiber sources; beverages. The visual Wheel of Five refers to this as fruits and vegetables; spreading and cooking fats; dairy, nuts, fish, legumes, meat, and eggs; bread, grain/cereal products, and potatoes; beverages. Each food group contains products that are beneficial to health or supply vital nutrients. It is advised to mainly eat the products that are included in the Wheel of Five, to eat a sufficient amount each day, and to vary between the food groups. The recommended dietary patterns are summarized in seven general recommendations for the Wheel of Five: (1) eat lots of fruit and vegetables; (2) consume mainly wholegrain products, such as wholegrain bread, wholegrain pasta, and brown rice; (3) eat less meat and more plant-based foods, and vary with fish, pulses, nuts, eggs, and vegetarian products; (4) consume sufficient dairy products, such as milk, yogurt, and cheese; (5) eat a handful of unsalted nuts daily; (6) consume soft and liquid spreadable fats and cooking fats; (7) drink sufficient amounts of tap water, tea, and coffee [14].

In 2019, the Dutch government announced that it intended to introduce the Nutri-Score as a national FOP label [15]. Nutri-Score is a FOP labeling system that classifies products from a green “A” to a red “E” based on their nutritional values [16]. The algorithm is based on giving negative points for energy, sugar, saturated fat, and salt content and positive points for fiber, protein, fruits, and the percentages of fruits, vegetables, pulses, nuts, rapeseed, walnuts, and olive oils. The sum of the positive and negative points determines the final classification. Foods with, for example, a higher salt content can be compensated to obtain a higher score by having, for example, a high protein content. The rewarding of nutrients and foods of which the consumption needs to be promoted seems to resonate well with the Dutch FBDGs. However, there is doubt whether the FOP Nutri-Score label in its current form is sufficiently aligned with the Dutch Wheel of Five [17,18]. We believe that this is due to the intrinsic design of the Nutri-Score methodology, i.e., one set of criteria for all pre-packaged foods, despite the criteria modifications that have been made specifically for cheeses, fats, and non-alcoholic drinks, because the scores of these products would not be in line with dietary recommendations. Therefore, we selected an alternative FOP labeling system, which can support a five-level graded FOP label and is product group-specific, i.e., with criteria specifically defined for each product group.

The Choices International Foundation (Choices) has developed the Choices criteria as a tool to improve a population’s diet and create a global standard for healthier food. It is a nutrient profiling system (NPS) that can support multiple food system actions, including FOP labeling. Choices categorizes food products in 1 of the 12 product categories, which can then be divided into 33 product groups. These product groups are divided into basic and non-basic product groups. Basic product groups contain essential nutrients, such as fruits, grains, meats, fish, and dairy. Non-basic product groups are discretionary products, which are not needed to stay healthy [19]. Each product group has specific criteria for salt, sugar, saturated fat, trans fat, energy, and fiber as thresholds that all the nutrients must meet in order to be compliant and eligible for an endorsement FOP logo. Recently, the Choices logo criteria have been extended to a five-level system, in which each food product receives a score ranging from 1 to 5 [20]. The Choices five-level criteria are the only known NPS criteria that are product group-specific and can be used to support a graded FOP labeling system, such as the Nutri-Score label. However, in contrast to Nutri-Score, the Choices NPS includes only fiber as a positive qualifying component, but fruits or vegetables, wholegrain, protein-rich foods, nuts, or specific oils are not taken into account. As the qualifying components and methodologies of the Choices and Nutri-Score NPSs are very different, in this paper, we investigate how well the Choices five-level criteria align with the Dutch FBDGs in comparison to Nutri-Score.

## 2. Materials and Methods

### 2.1. Data Collection

The products used in the comparison were selected from the Dutch Food Composition Database (NEVO) version 2019 [21]. This selection of products was made in such a way that the five food groups of the Wheel of Five were well represented in their diversity. The aim was to have a broad variety of products within each food group. For example, when looking at milk, skimmed milk, semi-skimmed milk, and whole milk were selected, and non-dairy milk substitutes and soft drinks were included in the beverages group. For products such as rice and other grain products, we chose to include the data for the unprepared food items (needed for Choices and Nutri-Score), but equivalent data for prepared food items (needed for the Wheel of Five) were available in NEVO as well. In the end, there were a total of 124 products selected (Appendix A).

### 2.2. Data Analysis

#### 2.2.1. The Wheel of Five

The Wheel of Five, shown in Figure 1, was used as a basis for the comparison. For each of the 124 selected products, compliance with the Wheel of Five was determined by using the guidelines for the Wheel of Five [22]. These guidelines provide detailed and quantitative criteria, as well as example products, to determine whether a food product complies with the Wheel of Five or not. The results can be found in Appendix A.

#### 2.2.2. Choices Score

The Choices score calculation was based on the five-level criteria that were published by Tognon and colleagues [20]. They defined criteria for the key nutrients: saturated fat, sodium, sugar, fiber, and energy for 33 product groups, divided into basic and non-basic product groups. To evaluate the products against the Choices criteria, the products were assigned to 1 of the 33 Choices product groups and the composition data from the NEVO database were compared with the Choices criteria for the product group. A calculation tool made by The Choices International Foundation was used. The product description, product group classification, whether it was a basic/non-basic product, the compositional data for the key nutrients, and the Choices score can be found in Appendix A.

#### 2.2.3. Nutri-Score

Nutri-Score only distinguishes between four product groups: cheeses, fats, beverages, and all other foods. To calculate the Nutri-Score, the products were divided into these product groups [23]. The percentages of fruit, vegetables, pulses, nuts, rapeseed, walnuts, and olive oils were required to calculate the Nutri-Score. As this information was not readily available in the NEVO database, the authors agreed to give selected products in the fruits, vegetables, pulses, and nut food groups, as well as olive oil a score of 100%, i.e., the maximum points for the content of these positive nutrients. The Nutri-Scores were calculated with the Belgium calculator tool [24]. The results can be seen in Appendix A.

### 2.3. Evaluation of the Product Selection

To assess whether the selection of products in this study was sufficient representative, we compared the Nutri-Score results of this study (see Section 2.2.3) with those of a study in which data from national food composition databases for >11,000 foods across 8 European countries were used to estimate the performance of Nutri-Score to discriminate the nutritional quality of products [25]. This was conducted by assessing the distribution of foods across the Nutri-Score classes within food groups. To compare the study of 8 European countries with the selection of products in this study, the food groups in the European study were combined in such a way that they were similar to the Wheel of Five food groups. Potatoes were included in the ‘bread, grain/cereal, and potatoes‘ food group and the pulses were included in the ‘dairy, nuts, fish, legumes, meat, and eggs’ food group. The results show that four Wheel of Five food groups were in line with the results of this study for Nutri-Score. Of the ‘spreading and cooking fats’, 18 + 64 = 82% were classified in C + D vs. 44 + 44 = 84% C + D in this study. The results for ‘fruit and vegetables’ were 86 + 9 = 95% A + B vs. 73 + 9 = 82% A + B, ‘dairy, nuts, fish, legumes, meat, and eggs’, 37 + 18 = 55% A + B vs. 38 + 25 = 63% A + B, and ‘beverages’, 24 + 16 = 40% A + B vs. 25 + 8 = 33% A + B. Differences were smaller than 15 percentage points. For the fifth food group, the results of the European study were slightly less in line with the present study: ‘bread, grain/cereal, and potatoes’ (33 + 17 = 50% A + B vs. 58 + 15 = 73% A + B). The difference was smaller than 25%. Overall, we concluded that the selection of products in this study represented a sufficiently broad variety of the different products within the five Wheel of Five food groups. Details can be found in Appendix A.

### 2.4. Data Processing

For evaluation against the criteria, all products were divided into the Wheel of Five food groups. The evaluation focused on a total of 124 products, all fitting with one of the Wheel of Five food groups. Choices and Nutri-Score both score products on a five-level scale: Choices assigns levels 1–5 and converts these levels from A to E (or 1–5) for basic food groups and C–E (or 3–5) for non-basic food groups, as per Figure 2 in the publication of Tognon and colleagues [20]. Nutri-Score normally calculates from A to E. In this comparison, Nutri-Score 1 equals Nutri-Score A, Nutri-Score 2 equals Nutri-Score B, etc. Levels 1–2 were considered to be healthy and, therefore, compliant with the Wheel of Five; levels 3–5 were considered less healthy and not compliant with the Wheel of Five. Correct assessments by the NPSs (Choices, or Nutri-Score) are defined as follows: the NPS scores a 1 or 2 for a product that is compliant with the Wheel of Five or the NPS scores a 3, 4, or 5 if the product is not compliant with the Wheel of Five. The alignment is then calculated as the percentage of correct assessments.

## 3. Results

### 3.1. Comparison of Choices and Nutri-Score with the Wheel of Five

In Figure 2, Figure 3, Figure 4, Figure 5 and Figure 6, the Choices score and Nutri-Score are displayed together with the products that comply/do not comply with the Wheel of Five. Compliance of the products with the Wheel of Five implies a Choices or Nutri-Score of 1 and 2. A score of 3, 4, or 5 indicates non-compliance with the Wheel of Five.

In the fruit and vegetable group (Figure 2), it can be seen that 45.5% of the products comply with the Wheel of Five, which is larger than Choices with 27.2% compliance (Choices score 1 plus Choices score 2). Nutri-Score considers 81.8% (Nutri-Score 1plus Nutri-Score 2) compliant with the Wheel of Five. Figure 3 shows the spreading and cooking fats group. Of these products, 66.7% are compliant with the Wheel of Five and 66.7% are considered compliant according to Choices, while 0.0% is considered compliant according to Nutri-Score. In the dairy, nuts, fish, legumes, meat, and eggs group (Figure 4), 56.3% are compliant with the Wheel of Five and 56.3% are compliant according to Choices. Nutri-Score considers 62.5% compliant with the Wheel of Five. Figure 5, containing bread, grain/cereal, and potatoes, shows 36.4% of the products compliant with the Wheel of Five. Choices considers 48.5% to be compliant and Nutri-Score considers 72.8% of the products compliant with the Wheel of Five. The last figure (Figure 6) represents data for the beverages group; 25.0% are compliant with the Wheel of Five, similar to the Choices score. Nutri-Score considers 33.3% to be compliant with the Wheel of Five. Overall, the Wheel of Five has a compliance of 46.8% of the total amount of products. Choices considers 46.8% of the total amount of products healthy and Nutri-Score 61.3%. Comparing the values for the different food groups, it is illustrated that Choices and the Wheel of Five are more similar, with differences of less than 20 percentage points for all food groups than Nutri-Score with differences of more than 35 percentage points for the food groups ‘fruits and vegetables (Figure 2), ‘cooking and spreading fats’ (Figure 3), and ‘bread, grain/cereals, and potatoes’ (Figure 5).

### 3.2. Alignment with the Wheel of Five

In Figure 7, the alignment of Choices and Nutri-Score with the Wheel of Five is shown for each food group. In the fruit and vegetable group, Choices is 72.7% in alignment with the Wheel of Five, while Nutri-Score is 54.5% in alignment with the Wheel of Five. The spreading and cooking fats food group illustrates that Choices is 100% in alignment with the Wheel of Five and 33.3% is in alignment with the Wheel of Five for Nutri-Score. Within the dairy, nuts, fish, legumes, meat, and eggs group, Choices is 70.8% in alignment and Nutri-Score is 68.8% in alignment with the Wheel of Five. Choices is 81.8% in alignment in the bread, grain/cereal, and potatoes group, while 63.6% are in alignment with the Wheel of Five according to Nutri-Score. The beverages group shows that all the products are in alignment with Choices, while 91.7% are in alignment with the Wheel of Five according to Nutri-Score. Comparing these values illustrates that Choices has a better alignment with the Wheel of Five than Nutri-Score. Choices has more correct assessments in all the food groups than Nutri-Score, in particular in the ‘spreading and cooking fats’ food group, where the difference between Choices and Nutri-Score is larger than 60 percentage points, but also in the ’fruits and vegetables’ and ‘spreading and cooking fats’ food groups, where the difference between Choices and Nutri-Score is 15 percentage points.

## 4. Discussion

Non-communicable diseases (NCDs) are among the top 10 leading causes of death [26]. Dietary choices can lead to, for example, being overweight, which increases one’s risk of these NCDs [27]. Therefore, it is important to encourage healthy diets and to make healthy choices easy choices [28]. Creating a clear FOP label to indicate the health of a product could support this but this should be aligned with national public health and nutrition policies and food regulations, including national FBDGs [3]. To evaluate this in the Dutch situation, a comparison between two NPSs (Choices five-level criteria and Nutri-Score), which both can be used to support a graded five-level FOP logo, was conducted to see which system fits better with the Dutch FBDGs, the Wheel of Five.

Results show that Choices is better aligned with the Wheel of Five than Nutri-Score for all five Wheel of Five food groups. The largest difference is in the spreading and cooking fats group (correct assessment percentages of 100% and 33%, respectively); differences in the fruit and vegetable group (73% and 55%) and bread, grain/cereal, and potatoes group (82% and 64%) are fairly large. An example is sunflower seed oil, which complies with the Wheel of five. Choices scores this product as a 1, in alignment with the Wheel of Five. However, Nutri-Score scores this product as a 4 or D. Another example, the product ‘pears in syrup tinned’, which does not comply with the Wheel of Five because of added sugar, is given a 4 by Choices, while Nutri-Score scores it as a 1 or A. This can be explained by the fact that high sugar levels can be compensated with fiber or fruit content. Smaller differences, less than 10 percentage points between Choices and Nutri-Score in alignment with the Wheel of Five, are found in the food groups ‘dairy, nuts, fish, legumes, meat, and eggs’ (correct assessment percentages of 71% and 69%, respectively) and ‘beverages’ with water, fruit juices, and soft drinks (100% and 92%). Results also indicate that Choices is stricter than Nutri-Score in all food groups except the spreading and cooking fats.

The Choices criteria do not align perfectly with the Wheel of Five. We discuss three examples. The selected dried fruits (dried figs, dried pear, and raisins) score a 5 following the Choices criteria, due to high total sugar content, but are included in the Wheel of Five (if no sugar is added). However, the NNC recommends limiting the consumption of dry fruits to 20 g/day. Another difference is that the Wheel of Five does not include any processed meat, whereas this is included in the Choices basic food groups and, therefore, can score a 1 or 2 if the criteria are met. Lastly, all (fresh, frozen, or processed) fish is included in the Wheel of Five with a recommendation to consume fish once a week. However, Choices has criteria for saturated fat and sodium for this product group, and some salted or smoked fish products did not meet the sodium criteria to be classified as healthy.

This is the first-known study that compares the alignment with the Wheel of Five between the Choices five-level criteria and Nutri-Score and illustrates that the Choices five-level criteria are better aligned with the Dutch Wheel of Five than the current Nutri-Score criteria. The findings of this study support the doubt about the suitability of the FOP label Nutri-Score in its current form. This is illustrated by the many discrepancies between the Nutri-Score and the Wheel of Five. It must be noted though that the Nutri-Score algorithm causing these discrepancies is currently under revision, and might improve in alignment with the Wheel of Five [29].

Two other studies have looked into the performance of Nutri-Score using food databases. The first was a comparative study across eight European countries on the performance of Nutri-Score to discriminate the nutritional quality of food products. The distribution of foods across the Nutri-Score classes within food groups was assessed. Data from national food composition databases for more than 11,000 foods were used [25]. In the methods section, this study was used to evaluate the selection of products used in the analysis of this study; it was concluded that the selection used here represented reasonably well the broad variety of the different products within the five Wheel of Five food groups. This evaluation can be found in Appendix A.

The second study was an evaluation of Nutri-Score with the Dutch Wheel of Five based on label data from more than 52,000 foods in the Dutch market [29]. The results show that for four Wheel of Five food groups, the distribution was reasonably well in line with the results of this study for Nutri-Score: ‘spreading and cooking fats’ (38 + 31 = 69% C + D vs. 44 + 44 = 88% C + D this study), ‘fruit and vegetables’ (89 + 4 = 93% A + B vs. 73 + 9 = 82% A + B), ‘bread, grain/cereal, and potatoes’ (37 + 28 = 65% A + B vs. 58 + 15 = 73% A + B), and ‘beverages’ (9 + 16 = 25% A + B vs. 25 + 8 = 33% A + B). For the food group ‘dairy, nuts, fish, legumes, meat, and eggs’ results were much less in line with the present study: (2 + 1 = 3% A + B vs. 38 + 25 = 63% A + B) [29]. Details can be found in Appendix A.

An explanation for the discrepancy between these two studies with Dutch data and European data from the 8 countries (excluding The Netherlands) could be the type of data that was used. As with the present study with Choices and Nutri-Score, the European evaluation with 8 countries used data from national food composition databases [25]. These data are more aggregated representing averages of more than one food product. While the label data in the second study represents a snapshot of all the food products in the market. This is a different type of food data, leading to different results. For example, the number of cheeses included in the Dutch study was very large (>4000), all scoring D or E. This influenced the outcome for the ‘dairy, nuts, fish, legumes, meat, and eggs’ food group, which resulted in a much lower percentage of products scoring for A + B. These results indicate that the outcome of an analysis such as this also depends on the type of data [29].

Our study comparing Choices and Nutri-Score with the Wheel of Five also has some limitations. First, the number of products included in this study was very limited. Comparisons of the results of this study and those with other studies based on data from more products show that, on the one hand, similar distributions were found for Nutri-Score when national food composition data were used [25,29]. On the other hand, with an even larger amount of branded label data, the results for Nutri-Score differed substantially [29]. This illustrates that the outcome was influenced by the choices of food products included in the analysis. We aimed for a good variation; however, it is advised to repeat this analysis with a larger sample of products. Secondly, industrial trans fatty acids (TFA) were originally part of the Choices criteria; however, the NEVO database does not discriminate between industrial TFA and ruminant TFA [21]. Since no distinction could be made, TFA was left out of the calculation. However, we believe that this simplification is justified for The Netherlands, as industrial TFA has been largely eliminated from processed foods [30].

Thirdly, the percentages of fruits, vegetables, pulses, nuts, rapeseed, walnuts, and olive oils were imputed since not all of the necessary information was available in the NEVO database. The fruits, vegetables, nuts, and olive oils all obtained a score of 100%. This way, these products are still taken into account, but it can also mean the given score is higher than if the percentage were specifically calculated. It is advised to calculate these percentages uniformly in future research. Moreover, it is recommended that researchers perform the comparison in different countries, comparing with other FBDGs, as we are not aware of other studies that have compared the alignment of the Choices five-level criteria with other FBDGs.

## 5. Conclusions

In summary, this study indicates that the Choices NPS is better aligned with the Wheel of Five than the current Nutri-Score NPS and that Choices is generally stricter than Nutri-Score. Whether the Choices five-level criteria is a suitable alternative for a FOP logo needs to be established in further studies.

## Figures and Tables

**Figure 1 nutrients-14-03527-f001:**
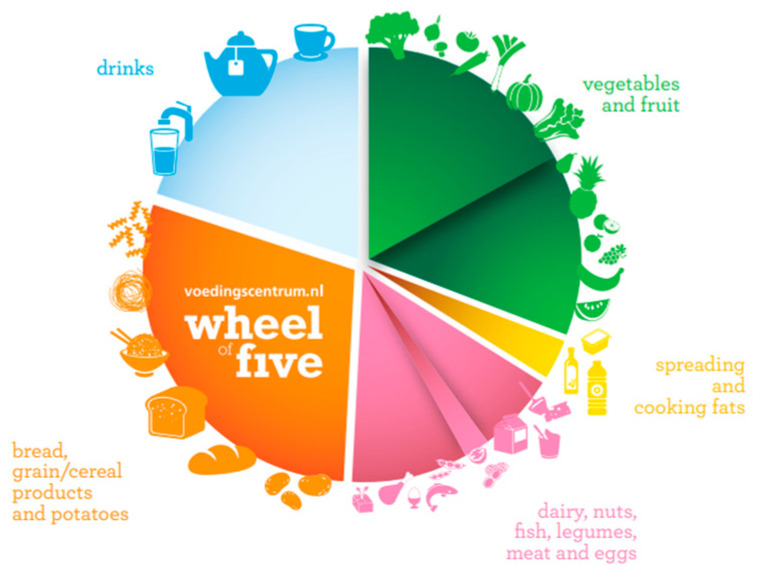
Wheel of Five of The Netherlands [14].

**Figure 2 nutrients-14-03527-f002:**
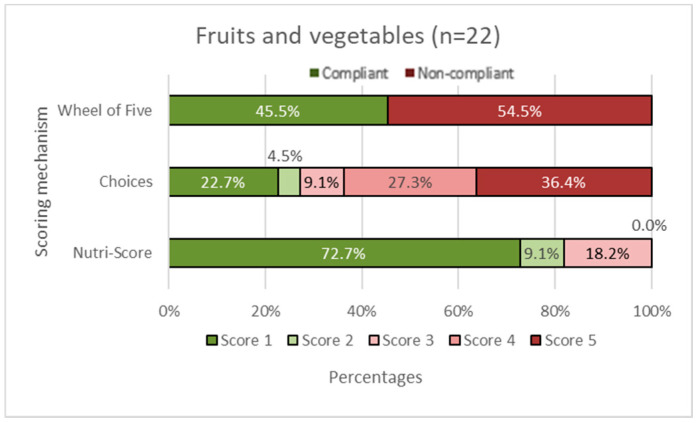
Fruit and vegetable group. The first horizontal bar shows the percentages of products that are compliant/non-compliant with the Wheel of Five. The Choices score and Nutri-Score show the percentages of products that scored 1 to 5. Nutri-Score 1 equals a score of A, Nutri-Score 2 equals a score of B, etc. Scores 1 and 2 (green shades) are considered healthy (compliant with the Wheel of Five) and scores 3, 4, and 5 (red shades) are considered less healthy (non-compliant with the Wheel of Five).

**Figure 3 nutrients-14-03527-f003:**
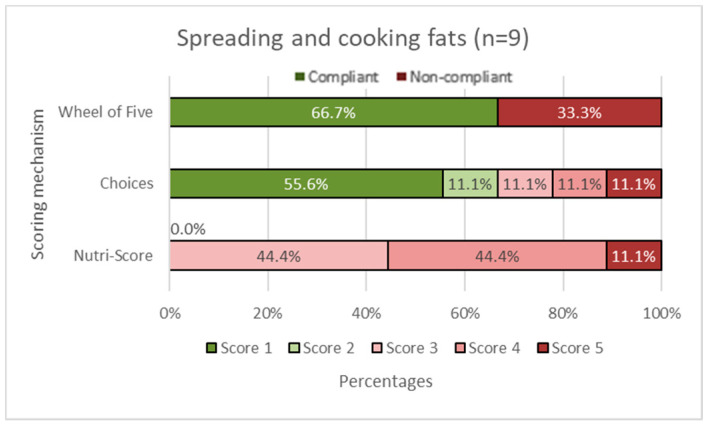
Spreading and cooking fats group. The first horizontal bar shows the percentages of products that are compliant/non-compliant with the Wheel of Five. The Choices score and Nutri-Score show the percentages of products that scored 1 to 5. Nutri-Score 1 equals a score of A, Nutri-Score 2 equals a score of B, etc. Scores 1 and 2 (green shades) are considered healthy (compliant with the Wheel of Five) and scores 3, 4, and 5 (red shades) are considered less healthy (non-compliant with the Wheel of Five).

**Figure 4 nutrients-14-03527-f004:**
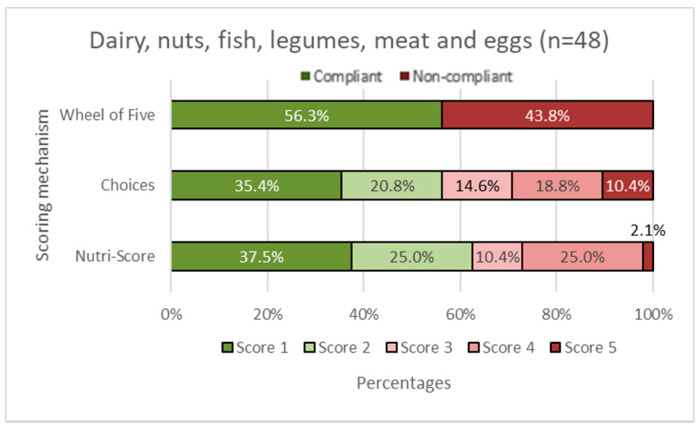
Dairy, nuts, fish, legumes, meat, and eggs group. The first horizontal bar shows the percentages of products that are compliant/non-compliant with the Wheel of Five. The Choices score and Nutri-Score show the percentages of products that scored 1 to 5. Nutri-Score 1 equals a score of A, Nutri-Score 2 equals a score of B, etc. Scores 1 and 2 (green shades) are considered healthy (compliant with the Wheel of Five) and scores 3, 4, and 5 (red shades) are considered less healthy (non-compliant with the Wheel of Five).

**Figure 5 nutrients-14-03527-f005:**
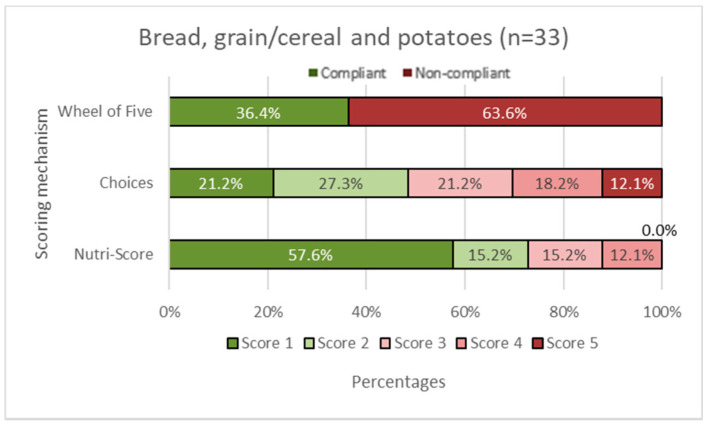
Bread, grain/cereal, and potatoes group. The first horizontal bar shows the percentages of products that are compliant/non-compliant with the Wheel of Five. The Choices score and Nutri-Score show the percentages of products that scored 1 to 5. Nutri-Score 1 equals a score of A, Nutri-Score 2 equals a score of B, etc. Scores 1 and 2 (green shades) are considered healthy (compliant with the Wheel of Five) and scores 3, 4, and 5 (red shades) are considered less healthy (non-compliant with the Wheel of Five).

**Figure 6 nutrients-14-03527-f006:**
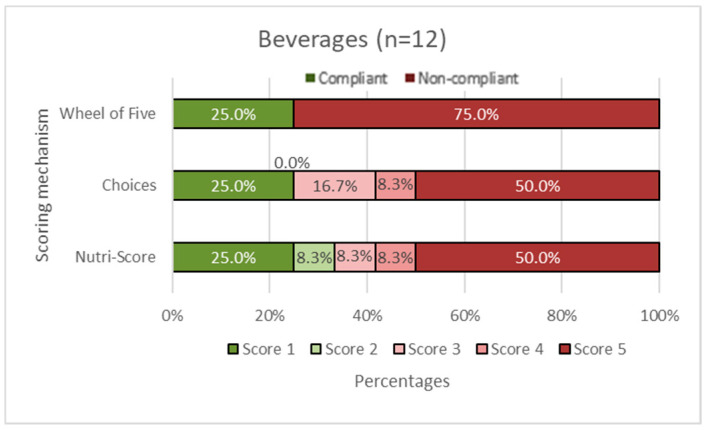
Beverages group. The first horizontal bar shows the percentages of products that are compliant/non-compliant with the Wheel of Five. The Choices score and Nutri-Score show the percentages of products that scored 1 to 5. Nutri-Score 1 equals a score of A, Nutri-Score 2 equals a score of B, etc. Scores 1 and 2 (green shades) are considered healthy (compliant with the Wheel of Five) and scores 3, 4, and 5 (red shades) are considered less healthy (non-compliant with the Wheel of Five).

**Figure 7 nutrients-14-03527-f007:**
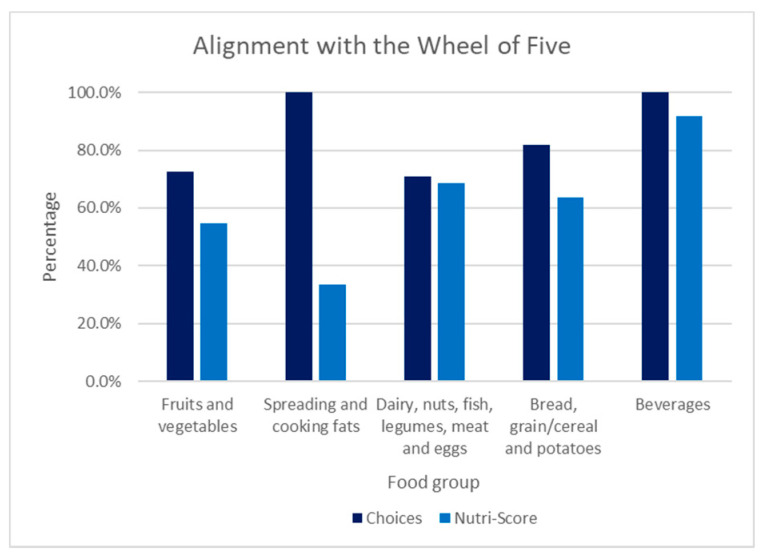
Extent of alignment of the NPSs (Choices and Nutri-Score) with the Wheel of Five (as percentages of correct assessments) for each food group of the Wheel of Five. A correct assessment is a score of 1 or 2 for a product compliant with the Wheel of Five or a score of 3, 4, or 5 for a product that is not compliant with the Wheel of Five.

## Data Availability

All additional data are included in the Appendix A.

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
