# Peer review of "Comparison between the Choices Five-Level Criteria and Nutri-Score: Alignment with the Dutch Food-Based Dietary Guidelines"

_nutrients, 2022, doi:10.3390/nu14173527_

Round 1
Reviewer 1 Report
This study addresses an important issue at a time when obesity risk is increasing, specifically the research considers how to align front of pack nutrient profiling systems with food-based dietary guidelines to support healthier choices.
General Recommendations
Overall, the paper is well written although it would benefit from some English editing to correct minor grammatical errors.
In considering what front of pack nutrient profiling system may be a good fit, it would be helpful in the Discussion to describe how the criteria for the Nutri-Score and Choices tools compare to Dutch government dietary guidance. It appears both tools consider energy, sugar, saturated fat, salt, and fiber—are these the primary Dutch dietary goals? What are other Dutch dietary goals and how could this impact selection of a tool? For example, from what is described in the manuscript, only Nutri-Score considers % of fruit/vegetables, pulses, and protein. Is increasing the % of fruit/vegetables, pulses, and protein a dietary goal/recommendation from the Dutch government? Adequate protein can be an important dietary goal/recommendation given that globally rates of sarcopenia are increasing, particularly for older populations.
The Discussion includes information about several Nutri-Score studies, is there research on the Choices tool and how it compares to dietary guidance? How does the present study compare to other research on the Choices tool?
Specific Recommendations
Line #66: how/why was Choices selected for this study vs. other FOP tools beyond Nutri-Score?
Line 252: What is the purpose of front of pack labeling? It seems a more nuanced discussion is needed here. If the purpose is to compare processed vs. non-processed foods, both tools seem to do this. If the purpose of these tools is also to compare the potential nutritional value among various processed foods, is this a benefit of Nutri-Score, i.e. do foods with higher fiber/fruit content offer higher nutritional value, even with higher sugar levels, given that fiber/fruit content may be contributing to the sugar/carbohydrate levels?
Line #268: suggest stating this is “one of the first studies” or the “first-known study”
Line 270: the statement “confirms the doubt of the suitability of the FOP label Nutri-Score” is very definitive, not sure that this single study of 124 products is definitive
Reviewer 2 Report
Comparison between the Choices 5-Level Criteria and Nutri-2 Score: alignment with the Dutch Food-Based Dietary Guidelines
Jaimy J. C. Konings, Herbert Smorenburg, and Annet J. C. Roodenburg
Summary: Authors were interested in exploring whether the Choice-5 level nutrient profiling system would be a better alternative to the Nutri-score front-of-package labeling system. The authors argue that the Nutri-score front-of-package labeling system may not align with the dietary guidelines set forth by the Dutch (called the Dutch Wheel of Five). The authors compared the Choice-5 level nutrient profiling system to the Nutri-score labeling system and found that Choices-5 had more overlap with the Wheel of Five and would be a better labeling system.
Overall, authors are attempting to explore an important area as it is key to ensuring the effectiveness of front-of-package labeling. The way authors set up the problem in the Introduction and summarized results, pointing to its implications in the Discussion, was sufficient. The primary concerns, however, are related to the Methods and their lack of clarity. Below, there are a few examples, but authors would benefit from completely revising the Methods section to ensure that any reader could follow their process and complete their calculations:
· It was very challenging to follow the authors’ descriptions of how they compared the nutrition profile systems and how they calculated the overlap.
· It is extremely difficult to follow the paragraph to evaluate the product selection. Please revise for more clarity. For example, what does it mean for a product to be 37+18% A+B vs. 38+25% A+B?
Additional comments:
Abstract
No comments.
Introduction
Please be sure to briefly review the other literature comparing nutrient systems.
Methods
Since the authors’ primary objective is to compare the Wheel of Five to the Choice-5 profiling system, it would be better to simply exclude those food items that could not be assessed using the Wheel of Five rather than using a different app (“KiesIkGezond?”)
When calculating the Nutri-score and deciding to give items a 0% or 100% if the NEVO database did not include sufficient information, authors will need to provide more information about their process for reaching consensus among the researchers
Results
Once authors clarify the Methods, it is suggested they revise the way they present the Results as well.
Discussion
No comments.
